# Effect of Stress on Quality of Life of Shift Nurses in Tertiary General Hospital: The Mediating Effect of Mindfulness

**DOI:** 10.3390/healthcare11010071

**Published:** 2022-12-26

**Authors:** Eunhee Hwang

**Affiliations:** Department of Nursing, Wonkwang University, Iksan 54538, Republic of Korea; ehh@wku.ac.kr; Tel.: +82-63-850-6071

**Keywords:** mindfulness, nurse, quality of life, stress

## Abstract

The purpose of this study was to examine the mediating effects of mindfulness on the relationships between stress and quality of life among shift nurses. A descriptive cross-sectional survey design was conducted using purposive sampling of 206 shift nurses in a tertiary general hospital in Korea. Data were analyzed with descriptive statistics, Pearson correlation and multiple regression analysis using SPSS/WIN 26.0 program. There were significant correlations among stress, mindfulness and quality of life. The quality of life had a positive correlation with mindfulness (r = 0.52, *p* < 0.001) and a negative correlation with stress among nurses. Mindfulness acts as a mediator in explaining relationship between stress and quality of life. This study provide evidence for the role of mindfulness in the relationship between stress and quality of life in shift nurses. Mindfulness appears to be a protective factor against nurses’ stress. If mindfulness-based interventions are developed and applied to improve the quality of life of shift nurses based on this study, it can help reduce their stress and improve the quality of life, which could ultimately improve the quality of nursing care for patients.

## 1. Introduction

Nurses play a very important factor in affecting patient outcomes and nursing outcomes [1,2]. In the recent nursing field, multi-dimensional and holistic nursing is required, which involves the expansion of the range of nursing recipients and physical, spiritual, mental, and social aspects. However, due to the lack of human and physical resources in the medical field, especially the lack of nursing personnel, the various needs and expectations of nursing subjects are not being met. Rather, nurses’ work intensity and stress are getting higher and higher.

One of the characteristics of nursing is shift work, which causes health problems such as physical and mental illness [3]. In fact, working shift causes work-related stress, burnout, high turnover, and safety accidents, and affects the quality of life [4,5,6,7]. Many countries across the world including Korea are experiencing a nursing shortage and a major factor contributing to the shortage is nurses working shifts. Thus, it is necessary to pay more attention to the physical and mental health of shift nurses.

Stress is often defined as an internal or external challenge, disorder, or stimulus; others perceive stress as a physiological challenge or response [8]. Nurses in Korea have a higher level of stress than ordinary people and professional workers [4]. Stress from working shifts can cause depression as well as various physical symptoms, and result in high rates of absenteeism and turnover caused by illness or injury [9,10,11,12,13]. Quality of life is a subjective state of well-being felt by an individual in the physical, mental, and social domains and is an index that represents an individual’s overall well-being [14]. Quality of life, which is inherent in physical, cultural, and social contexts, is one of the most important aspects of human health [15]. Several studies have found that stress, burnout, and negative emotions felt in daily life are related to nurses’ quality of life and that quality of life directly or indirectly affects work efficiency, quality of nursing care, job satisfaction, and intention to turnover [16,17,18]. In other words, the quality of life of nurses is a factor that affects not only the nursing but also the personal aspect.

Work context including stress inevitably occurs in the nursing work environment and consequently contributes to the quality of life [19]. Quality of life is a predictor of nurses’ turnover intention [20]. Considering the effect of quality of life on nursing work, efforts to improve quality of life are necessary, but many studies have focused on reducing the factors that affect quality of life. However, it is not easy to prevent nurses from being stressed in the increasingly complex clinical field. Therefore, a new approach that can prevent negative effects from nursing work is needed, which will be a more effective way to improve the quality of nursing care.

Mindfulness, a crucial psychological factor, is an emerging concept in positive psychology and refers to increased attention to open and receptive awareness of one’s current experience or reality [21]. By clearly noticing one’s reactions, mindfulness allows individuals to live their lives wisely, getting away from unconscious and habitual reactions, so that individuals can effectively respond to unavoidable negative situations [22]. In the increasingly diverse and complex clinical field, causes of stress and stress levels will increase. In this situation, it is necessary to focus on removing or changing nurses’ stress-inducing factors, while exploring ways to strengthen the individual’s capacity to help overcome stress. Mindfulness can be considered a way for nurses to reduce the negative effects of stress they experience. Previous studies have shown that mindfulness reduces stress and burnout. [23,24]. In addition, systematic reviews of healthcare providers reveal that practitioners who incorporate mindfulness into their personal and professional lives demonstrate an improved sense of well-being and ability to employ self-care strategies [25,26]. 

This study formulated a conceptual framework for variables as follows based on previous studies to identify the mediating effect of mindfulness in the relationship between nurses’ stress and quality of life. A study of Chinese surgical nurses [16] found that high levels of stress were associated with low quality of life. Chronic stress makes it difficult for nurses to control their emotions. Furthermore, a negative attitude toward work and life, indifference toward the recipient, and loss of emotion lead to burnout [27,28], which negatively affects the quality of life and care [29]. A study of employees at large companies found that mindfulness is a stress-reducing factor [30], and a study of clinical nurses [4] also proved that mindfulness helps reduce stress. Based on the fact that mindfulness influences stress and stress affects the quality of life, a conceptual framework that mindfulness plays as a mediating factor in the relationship between stress and quality of life was established. 

The purpose of this study is to identify the correlation between stress, mindfulness, and quality of life and to verify the effect of mindfulness as a parameter to develop a quality of life improvement program for shift nurses. The specific purposes of this study are as follows.

Identify the general characteristics, the level of mindfulness, stress, and quality of life of the participants.Identify differences in levels of quality of life according to the general characteristics of the participants.Identify the correlation between mindfulness, stress, and quality of life.Verify the mediating effect of mindfulness in the relationship between stress and quality of life.

## 2. Materials and Methods

### 2.1. Design

The descriptive cross-sectional survey design was conducted using purposive sampling of 206 shift nurses in a tertiary general hospital in Korea.

### 2.2. Participants

The participants of this study were shift nurses working in a tertiary general hospital in Korea. When a significance level of 0.05, a medium effect size of 0.15, a power of 0.95, and 11 predictors were set in the regression analysis using G*power 3.1.9.2 [31], the number of samples required was 178 in total. This study selected a total of 215 people considering the dropout rate and analyzed a total of 206 people, excluding 9 people who gave insufficient responses. 

### 2.3. Measurements

#### 2.3.1. Mindfulness

Mindfulness was measured using the mindfulness scale developed by Park [32]. This scale is composed of 4 sub-factors: present awareness, attention, non-judgmental acceptance, and decentralized attention, with a total of 20 items, 5 items for each factor. Present awareness is the clear and immediate awareness of the experiences taking place in the body and mind in the present moment, and attention measures the extent to which attention is focused and maintained on a current experience or task. Non-judgmental acceptance measures the attitude of accepting and helps embrace experiences as they are without subjective assessment or judgment of one’s inner experiences. Decentralized attention refers to the power of looking at a situation as an observer at a distance without being too preoccupied with how they feel. Each item was measured on a 5-point Likert scale ranging from 1 point for “not at all” to 5 points for “very much so”. All were reverse-scored, with higher scores indicating higher levels of mindfulness. In the tool development stage, Cronbach’s α value was 0.88 in Park [32] and Cronbach’s α value was 0.94 in this study.

#### 2.3.2. Stress

For stress, this study adopted the Korean version of the perceived stress scale (PSS) with a 10-item questionnaire originally developed by Cohen et al. [33] and revised in 1988, which was later translated by Lee Jong-ha et al. [34]. Items 1, 2, 3, 6, 9, and 10 are positively worded items (0 = never, 1 = rarely, 2 = sometimes, 4 = very often), and negatively worded items (Items 4, 5, 7, 8) were reverse scored, which indicates that the higher the score, the greater the perceived stress level. In the study conducted by Lee et al. [34], Cronbach’s α value was 0.82, and in this study, Cronbach’s α value was 0.82.

#### 2.3.3. Quality of Life

As for the quality of life, this study used the Scale for Korean Adults’ Quality of Life (SKAQOL) tool, which was developed by No [35] and later modified and supplemented by Park [36]. The sub-domains of this tool include 28 items in total: self-esteem (5 items), working conditions (7 items), leisure activities (5 items), emotional state (6 items), physical condition and function (2 items), and family and friend relationships (3 items). Each item is on a 5-point Likert scale ranging from 5 points for ‘very much so’ to 1 point for ‘not at all, and negatively worded items were reverse scored. Higher scores mean a higher quality of life. Cronbach’s α value was 0.89 in the study of Park [36], and it was 0.89 in this study.

### 2.4. Data Analysis

The SPSS/WIN 26.0 program was used to analyze the collected data. General characteristics were calculated as percentage, mean, and standard deviation, and independent t-test, ANOVA, Tukey test, and Pearson correlation coefficient were used to analyze the relationship between general characteristics and mindfulness, stress, and quality of life. To diagnose whether the data of this study are suitable for regression analysis, this study tested the assumptions of regression analysis. The Durbin-Watson statistic ranged between 1.662 and 1.822, the tolerance limit ranges between 0.670 and 1, and the variance inflation factor ranges between 1 and 1.494. These tests confirmed equal variance in the residual plot, verifying the assumptions of independence of the residuals, normal distribution, an equal variance of the dependent variable, and multicollinearity.

This study validated the model according to the mediating effect analysis method of Baron et al. [37]. With mindfulness as a parameter, step 1 confirms whether the independent variable, stress, is statistically significant in the parameter, mindfulness level, and step 2 verified the independent variable, stress, is statistically significant in the dependent variable, quality of life. In the third step, the independent variable, stress, and the parameter, mindfulness, were simultaneously put into the regression equation to verify the effects of stress and mindfulness on the quality of life, respectively. To verify the significance of the mediating effect of mindfulness, the Sobel test was conducted using the standard error between unstandardized coefficients [38].

### 2.5. Ethical Considerations and Data Collection

This study was conducted after receiving IRB approval (WKIRB-202110-SB-083) from the bioethics review committee of the university of which the research director is part with the aim of carrying out ethical research. Prior to data collection, the purpose and method of the study, anonymity, and confidentiality were explained to the administrator of the institution, and permission was obtained before data collection. Data were collected online using Google Surveys. Subjects provided informed consent to the purpose of research, collection of personal information, and the principle of confidentiality on the first page of the survey and participated voluntarily. In addition, the survey results were automatically processed through a system after the completion of the survey for anonymization. Data were collected from 1 December 2021 to 21 December 2021.

## 3. Results

### 3.1. General Characteristics

The average age of the participants was 31.79 ± 7.01 years, 195 (94.7%) were female and 11 (5.3%) were male. As for positions, 192 (93.2%) were staff nurses and 14 (6.8%) were charge nurses. As for departments, 139 (67.5%) work in general wards, 51 (24.8%) in intensive care units, and 16 (7.8%) in emergency rooms and others. The total clinical experience was 103.69 ± 84.21 months, and the period worked at the current department was 39.84 ± 42.07 months. In addition, there was no significant difference in the quality of life of the subjects according to general characteristics (Table 1).

### 3.2. Level of Mindfulness, Stress, and Quality of Life

The participants’ mindfulness levels were an average of 3.47 ± 0.67 points (out of 5 points), stress levels were an average of 1.92 ± 0.48 points (out of 4 points), and levels of quality of life were an average of 2.85 ± 0.46 points (out of 5 points) (Table 2).

### 3.3. Correlation between Mindfulness, Stress, and Quality of Life

Table 3 shows the results of the correlation analysis among mindfulness, stress, and quality of life. The subjects’ quality of life had a positive correlation with mindfulness (r = 0.52, *p* < 0.001) and a negative correlation with stress (r = −0.64, *p* < 0.001). In other words, the higher the mindfulness, the higher the quality of life, and the higher the stress, the lower the quality of life.

### 3.4. Mediating Effects of Mindfulness between Stress and Quality of Life

According to the procedure suggested by Baron et al. [37], the model was analyzed to verify the mediating effect of mindfulness on the relationship between stress and quality of life. In step 1, the independent variable, stress, had a significant negative correlation with the parameter, mindfulness (β = −0.58, *p* < 0.001). In step 2, the independent variable, stress, showed a significant negative correlation with the dependent variable, quality of life (β = −0.64, *p* < 0.001), and in the last step, both the independent variable, stress, and the parameter, mindfulness, were put into the regression model. Mindfulness, the parameter, was found to have a significant effect on the quality of life (β = 0.23, *p* < 0.001), and stress, an independent variable, also had a significant effect on the quality of life. Its influence was higher than that of the second step (β = −0.64→β = −0.51). In other words, mindfulness had a partial mediating effect on the relationship between stress and quality of life. The explanatory power that stress has over quality of life was 40.6% (F = 141.18), and the explanatory power that stress and mindfulness have over the quality of life increased to 43.9% (F = 81.11), indicating that mindfulness influences quality of life. (Table 4, Figure 1). To verify the significance of these mediating effects, the Sobel test was conducted. If the Z value is larger or smaller than 1.96, the mediating effect can be considered significant. In this study, the Z value was −3.48 (*p* = 0.001), proving that there is a partial mediating effect of mindfulness in the relationship between stress and quality of life.

## 4. Discussion

This descriptive survey study aims to identify the correlation between stress, mindfulness, and quality of life and to verify the mediating effect of mindfulness.

The quality of life of the subjects was 2.85 out of 5 points. Compared to the average quality of life score of Korean tertiary general hospital nurses of 3.32 points [39] and that of Korean general hospital nurses of 3.27 points [40], the average score of the subjects is lower. It is also lower than the average score of Mexican nurses of 3.50 [41], which was originally 207.31 out of a total of 269 and converted into a 5-point scale. The participants in this study are shift nurses working at a tertiary general hospital, which aims to efficiently utilize medical resources by providing high-quality medical services for patients with severe diseases and establishing a medical delivery system [42]. Therefore, it is safe to assume that the amount and intensity of work of the subjects could be greater than that of nurses working in general hospitals. In addition, since the data were collected in December 2021 when the COVID-19 pandemic reached its peak, fatigue, burnout, and anxiety of the nurses caused by the infectious disease have accumulated for the last two years, resulting in a decrease in the quality of life. In this study, among the sub-domains of quality of life, physical condition and function scored the lowest with 2.42 points, which can be seen as reflecting the circumstances. Therefore, it is necessary to prepare a plan to strengthen the capacity of nurses to overcome any negative situation.

The average stress score of the subjects was 1.92 out of 4 points, which is higher than that of Korean general hospital nurses at 1.06 points [4], which was originally 41.45 (out of 156 points) and converted into a 4-point scale. Additionally, it is similar to the stress score of the Chinese tertiary general hospital nurses of 1.94 [43], which was 27.12 points (out of 56 points) and later converted into a 4-point scale. The study by Xie et al. [43] has a similar background as this study, targeting nurses working at tertiary general hospitals and carrying out data collection during the pandemic. This shows that nurses’ stress is affected by their work.

The average score for mindfulness of the subjects was 3.47 out of 5 points, which is similar to that of university hospital nurses at 3.50 [44], but lower than that of workers in large corporations at 4.01 [30]. Working shift is a factor causing burnout, work stress, and somatic symptoms in nurses [45]. The difference between this study and previous studies can be attributed to the fact that hospital nurses who work shifts have relatively little room for mindfulness compared to other full-time workers who do not. Mindfulness is an important mechanism to help an individual calmly accept when things get difficult and embrace unpleasant experiences. Thus, improving mindfulness helps develop the individual’s adaptive coping strategy, protect an individual from negative situations, and adapt to change [46]. Mindfulness enables shift nurses to effectively cope with various clinical situations and can be an effective intervention to promote mental health.

This study showed both stress and mindfulness have an effect on the quality of life, and mindfulness has a partial mediating effect in the relationship between stress and quality of life. This result confirms that stress not only directly affects the quality of life but also indirectly affects it through mindfulness. Therefore, the intervention of mindfulness is necessary as a strategy to relieve stress to improve quality of life. Mindfulness has a significant mediating effect on the relationship between various variables, such as the relationship between self-esteem and burnout in clinical nurses [44], the relationship between emotional intelligence and stress in clinical nurses [4], and the relationship between stress and quality of life in cancer patients [24]. These findings suggest that mindfulness helps to strengthen positive variables and alleviate negative variables. 

The study systematically examined nurses’ mindfulness and confirmed that mindfulness-based interventions had significant effects on emotional problems such as depression and anxiety and also confirmed that it benefits individual well-being and improves work performance (better communication with colleagues and patients, higher sensitivity to patients’ experiences, clearer analysis of complex situations, and emotional regulation in stressful contexts) [47]. Currently, studies are being conducted to explore the effectiveness of applying various intervention programs based on mindfulness in clinical settings, but there is a limited amount in Korea. However, in other countries, mindfulness-based intervention programs are being applied to improve various factors such as empathy fatigue, exhaustion, work errors in nursing, mental pain, and empathy as well as stress [48]. Evidence suggests that mindfulness is supportive of a variety of clinical populations, including but not limited to, the improvement of chronic and cancer-related pain, sleep disorders, eating disorders, psoriasis, and a variety of psychological disorders [49]. Nursing administrators in hospitals should identify the stress factors of shift nurses and try to improve the negative effects of these on nursing. They can use mindfulness-based interventions in supportive policies with the aim of reducing stress and improving quality of life of shift nurses. These policy strategies will ultimately improve the quality of nursing. 

Since this study has a limitation in that it did not examine the causal relationship between variables through a cross-sectional study, future studies with longitudinal design and mindfulness-based interventions will be needed. In addition, since the participants in this study were nurses working shift at a tertiary general hospital, repeated research design targeting health professionals in various clinical settings are needed to generalize the results of the study.

## 5. Conclusions

The study findings showed the stress levels of shift nurses are high, and the levels of mindfulness and quality of life are low. It also verified the higher the mindfulness of the subject, the higher the quality of life, and the higher the stress, the lower the quality of life. In addition, both stress and mindfulness were found to affect the quality of life, and mindfulness was found to have a partial mediating effect in the relationship between stress and quality of life. Reflecting on the trends of the times that value personal well-being, the quality of life is an important factor in choosing a job. Therefore, the quality of life of nurses also affects nursing outcomes such as turnover and retention. If mindfulness-based interventions are developed and applied to improve the quality of life of shift nurses based on this study, it can help reduce their stress and improve the quality of life, which could ultimately improve the quality of nursing care for patients.

## Figures and Tables

**Figure 1 healthcare-11-00071-f001:**
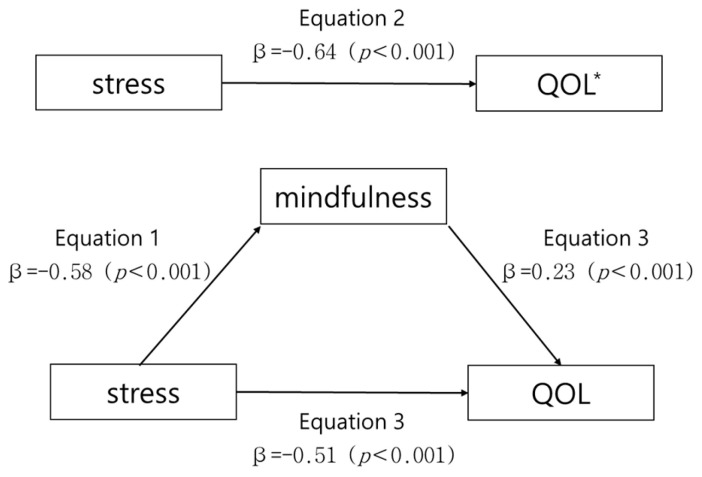
Mediating effect of mindfulness in the relationship between stress and quality of life. * QOL: Quality of life.

**Table 1 healthcare-11-00071-t001:** General characteristics (*N* = 206).

Characters	Categories	*n*(%)	Mean ± SD	t or F	*p*
Gender	Female	195(94.7)	2.84 ± 0.45	1.29	0.200
	Male	11(5.3)	3.03 ± 0.65		
Marital state	Single	131(63.6)	2.83 ± 2.83	−0.94	0.346
	Married	75(36.4)	2.89 ± 2.89		
Education	Associate degree	26(12.6)	2.86 ± 0.42	0.96	0.384
	Bachelor’s degree	157(76.2)	2.83 ± 0.44		
	≥Master’s degree	23(11.2)	2.98 ± 0.58		
Monthly income (USD)	Less than 2500	45(21.8)	2.83 ± 0.42	1.23	0.300
	2500-less than 3300	74(35.9)	2.82 ± 0.47		
	3300-less than 4100	18(8.7)	2.80 ± 0.54		
	4100-less than 5000	15(7.3)	2.96 ± 0.52		
	Over 5000	54(26.2)	2.93 ± 0.42		
Position	Staff nurse	192(93.2)	2.85 ± 0.47	−0.05	0.958
	Charge nurse	14(6.8)	2.86 ± 0.35		
Department	General ward	139(67.5)	2.84 ± 0.47	0.83	0.439
	Intensive Care Unit	51(24.8)	2.92 ± 0.44		
	Emergency Room, others	16(7.8)	2.79 ± 0.42		
Age(year)			31.79 ± 7.01		
Total period of clinical experience (Month)			103.69 ± 84.21		
Total period of current department experience (Month)			39.84 ± 42.07		

**Table 2 healthcare-11-00071-t002:** Level of mindfulness, stress, and quality of life (*N* = 206).

Variables	Min	Max	Mean	SD
Mindfulness	1.55	5.00	3.47	0.67
Attention	1.80	5.00	3.45	0.71
Decentralized attention	1.00	5.00	3.14	0.87
Non-judgmental acceptance	1.40	5.00	3.65	0.78
Awareness of present time	1.60	5.00	3.64	0.74
Stress	0.20	3.70	1.92	0.48
Quality of life	1.64	4.04	2.85	0.46
Self-esteem	2.20	5.00	3.45	0.58
Family and friend relationships	1.33	5.00	3.28	0.61
Leisure activities	1.00	4.60	2.78	0.72
Emotional state	1.00	4.50	2.82	0.68
Physical condition and function	1.00	4.00	2.42	0.65
Working conditions	1.14	4.14	2.45	0.63

**Table 3 healthcare-11-00071-t003:** Correlation among mindfulness, stress, and quality of life (*N* = 206).

Variables	Mindfulness	Stress
	R (*p*)	
Quality of life	0.52 (<0.001)	−0.64 (<0.001)
Mindfulness	1	−0.58 (<0.001)

**Table 4 healthcare-11-00071-t004:** Mediating effect of mindfulness (*N* = 206).

Step	Direction	B	SE	β	t	*p*	Adj. R^2^	F	*p*
Step1	Stress→Mindfulness	−0.81	0.08	−0.58	−10.03	<0.001	0.327	100.684	<0.001
Step2	Stress→QOL	−0.61	0.05	−0.64	−11.88	<0.001	0.406	141.18	<0.001
Step3	Stress→QOL	−0.49	0.06	−0.51	−7.94	<0.001	0.439	81.11	<0.001
	Mindfulness→QOL	0.16	0.04	0.23	3.58	<0.001			
		Sobel test Z = −3.48, *p* = 0.001							

QOL: Quality of life.

## Data Availability

The data presented in this study are available on request from the corresponding author. The data are not publicly available due to participants’ privacy.

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
