# Peer review of "Effect of Stress on Quality of Life of Shift Nurses in Tertiary General Hospital: The Mediating Effect of Mindfulness"

_healthcare, 2022, doi:10.3390/healthcare11010071_

Round 1

Reviewer 1 Report

This is an exciting topic about the stress nurses on shifts may face, affecting their quality of life. This study examines the mediating effects of mindfulness on the relationships between stress and quality of life among shift nurses, which is the main objective of this study. There are several excellent points in this manuscript. However, I have some issues that need some deep consideration.

1- To separate the primary and secondary titles, the research title should be revised to use ‘:’ instead of ‘;’.

2- The abstract is well written, describing the research focus, methods, and results yielded from the techniques used.

3 - The introduction needs more explanation of the literacy gap, the added value of the present research, and the contribution it makes to the field. Some recent references need to be included when describing.

“This study formulated a conceptual framework for variables based on previous studies to identify the mediating effect of mindfulness in the relationship between nurses’ stress and quality of life.” Please add these previous studies.

4- The research design should be clarified in terms of:

  • On Page 3, Line 118, there is a reference to the Perceived Stress Scale. I suggest adding a reference to this scale in the Korean context or mentioning the main questions used to test the sample’s stress level.
  • The criteria for selecting the sample size are mentioned in the abstract. References should accompany the equation for determining sample size. Double-check some connections that use the equation based on the random selection. This article could help determine the sample size [Workers’ Satisfaction vis-à-vis Environmental and Socio-Morphological Aspects of Sustainability and Decent Work]. Further references should also be mentioned.

5- The results and discussion were well-written and clearly showed the main findings. The debate shows the author’s conclusions by linking the main findings to previous studies. The limitations were also well explained. In future research, I would recommend examining the effects of other stress scales worldwide.

Author Response

Please see the attachment. Thank you for your detailed review.

Reviewer 2 Report

Introduction: I would suggestion this: In the affecting patient and nursing outcomes. 

Design: Indicate what kind of study : a descriptive cross-sectional 8 survey design was conducted using purposive sampling of 206 shift nurses in a tertiary general hospital in Korea.   

Participants: you could talk about the recruitment strategy. How did you recruitment the participants. 

Any implications for nursing on how to improve the quality of life of shift nurses ? giving some examples before the conclusion. 

Conclusion: what are the prospects for the future regarding retention. It would be great to do some link with the actual world retention nursing crisis. 

Author Response

(The authors gave the same response as above.)

Reviewer 3 Report

First of all, congratulations for the work done, then I will mention a number of changes and recommendations in order to obtain clearer and more accurate information.

- Comments on the introduction:

Lines 63-86: a grammatical improvement is needed.

Several sentences are very long and difficult to understand and read fluently.

- Comments on results:

If there is information in a table, it is not necessary to repeat the same information in the text.

I don't understand, if it is a Likert scale 1-5 how can the minimum be 1.80 in attention for example? (Table 2)

- Comments on discussion:

Your study is a descriptive study, then you cannot talk about the effects of mindfulness-based interventions on the QoL, stress...

Author Response

(The authors gave the same response as above.)

Round 2

Reviewer 1 Report

All my comments have been addressed and the authors provided proper answer. 

Author Response

Thank you for your comment, Please see the attachment.

Reviewer 3 Report

I am glad to see that most of the suggestions made by myself and the other reviewers have been taken on board and corrected, but I am still concerned about one issue:

Point 3: I don't understand, if it is a Likert scale 1-5 how can the minimum be 1.80 in attention for example? (Table 2)

Response 3: (Page 5) The minimum and maximum values presented in the table 2 indicate the range of the average distribution of the variables of the participants.

Again, I still do not understand that with Likert scale data (1-5), you can get a minimum value that is decimal. The minimum and maximum values are the minimum and maximum values of the results obtained by the subjects, therefore, if we have a Likert scale with whole numbers 1-5, the minimum and maximum must be whole numbers 1-5.

Author Response

(The authors gave the same response as above.)
